# Physical Fitness and Performance in Talented & Untalented Young Chinese Soccer Players

**DOI:** 10.3390/healthcare10010098

**Published:** 2022-01-04

**Authors:** Alfredo Irurtia, Víctor M. Torres-Mestre, Álex Cebrián-Ponce, Marta Carrasco-Marginet, Albert Altarriba-Bartés, Marc Vives-Usón, Francesc Cos, Jorge Castizo-Olier

**Affiliations:** 1INEFC-Barcelona Sports Sciences Research Group, National Institute of Physical Education of Catalonia (INEFC), University of Barcelona, 08038 Barcelona, Spain; vmtm3@yahoo.es (V.M.T.-M.); acebrian@gencat.cat (Á.C.-P.); mcarrascom@gencat.cat (M.C.-M.); mvivesu@gencat.cat (M.V.-U.); cosfrancesc@gmail.com (F.C.); 2Catalan School of Kinanthropometry, National Institute of Physical Education of Catalonia (INEFC), University of Barcelona (UB), 08038 Barcelona, Spain; 3Sport and Physical Activity Studies Center (CEEAF), Sport Performance Analysis Research Group (SPARG), University of Vic-Central University of Catalonia, Vic, 08500 Barcelona, Spain; albert.altarriba@uvic.cat; 4School of Health Sciences, TecnoCampus, Pompeu Fabra University, Mataró, 08302 Barcelona, Spain; jcastizo@tecnocampus.cat

**Keywords:** training, long-term, talent identification, team sport, multivariate analysis

## Abstract

Sports performance is a complex process that involves many factors, including ethnic and racial differences. China’s youth soccer is in a process of constant development, although information about the characteristics of its players and their methodological systems is scarce. The aim of this retrospective study was to characterize the physical fitness and the competitive performance of 722 Chinese players of three sports categories (8.0–9.9, 10.0–11.9 and 12.0–13.9 years), who were classified by their coaches as talented (*n* = 204) or untalented (*n* = 518). Players were assessed for anthropometry (body height, body mass, body mass index), lung capacity (Forced Vital Capacity), jumping performance (Squat Jump, Countermovement Jump and Abalakov tests), sprinting performance (10 m and 30 m Sprint tests), agility performance (Repeated Side-Step test) and flexibility (Sit & Reach test). A descriptive, comparative, correlational and multivariate analysis was performed. Competitive ranking was created in order to act as dependent variable in multiple linear regression analysis. Results indicate that Chinese players classified as talented have better motor performance than untalented ones. However, these differences are neither related nor determine the competitive performance of one group or the other.

## 1. Introduction

The People’s Republic of China (PRC) has a policy interest in promoting and developing professional soccer [1,2]. However, despite being one of the world’s major economic powers, PRC has had to establish pacts and alliances with European clubs in order to replicate its methodology and sports training systems [3,4]. Indeed, China currently lacks basic structures, and clubs are filled with professional soccer schools, which are generally advised by and/or linked to big European sports clubs, dedicated to developing and promoting possible future talented players [5]. Children share studies, social life, and sports training under increasingly well-equipped academic-sports structures. In return, European clubs reserve the right to select the best promising players and allow them to compete in their teams in the future [3,5].

Identifying sports talent is an exciting and complex area of sports science, due to its multidimensional nature [6,7,8,9,10]. Moreover, this complexity increases when competitive performance must be explained or predicted in a team sport such as soccer [11,12]. Cognitive constructs related to the greater or lesser success in the execution of certain tactical skills, performed by a player who always interacts with his teammates and opponents, are difficult to assess individually [13]. Thus, on many occasions, the criteria of expert coaches are applied to determine a general ranking of players based on a score [14,15]. Coach-based skill ratings have been adopted in previous research with demonstrated validity and reliability [6,16]. In relation to youth soccer, Hendry and colleagues [15] used a five-point scale (one = poor to five = excellent) to rate 102 elite youth players, according to their skill level. Another example is the contribution of Jukic and colleagues [14], who created a questionnaire for coaches with nine items that attempted to cover, multidimensionally, the quality they attributed to each player, according to a series of technical, tactical, physical, and psychological aspects. Each player was classified into four scores A = above average performance; B = average performance; C = low performance; D = below standards (the analysis model explained 58.49% of the variance).

If sports performance is taken as a continuous variable, multiple linear regression (MLR) is the most common statistical technique. Thus, this ranking determines the competitive performance and conforms the dependent or predicted variable to confront the different independent or predictive variables (e.g., anthropometric, physiological, technical skills, and/or physical fitness tests) when a multivariate statistical model is performed [17]. On the other hand, in connection with the independent variables, although no clear evidence has been found to support the use of physical skill or physiological tests in early stages of sports development to predict future success, its administration is an established process in all training programs, both in individual or team sports [18,19]. In this regard, a recent review on the identification of talented players in soccer highlighted the importance of evaluating different skills, scaling the children by age groups, and adding information on their anthropometric and physiological profiles [20]. However, although it is well characterized that young, successful soccer players have anthropometric and physical fitness characteristics that are differentiated from the rest of their peers [21,22], it is necessary to understand that it is unlikely that these differences provide a reliable source to predict success within an already talented group [23]. That is why it is recommended to use the collected data to guide the training programs, while considering its substantial limitations to predict future sport success [24,25].

Access to information on the methods applied for the identification of sports talent in the PRC and on the training processes of its young athletes is difficult. Language, culture, and the country’s own socio-political reality limit its availability. In soccer, there are very few scientific contributions published in English about the anthropometric profile, motor skills, or physiological characteristics of children and/or adolescents [26,27,28,29]. Therefore, the primary aim of this study was to provide information about anthropometric characteristics and performance in some physical fitness tests commonly used in an important Chinese soccer academy. Likewise, to help their managers and all technical staff, the possible relationships between the results of each test and a ranking established by the number of competitive victories was analyzed according to the competitive category and performance level (talented and untalented teams). Finally, from a multidimensional perspective, a multivariate analysis was performed to find out if any of these tests, or a set of them, could explain the variance of the ranking.

## 2. Materials and Methods

### 2.1. Study Design

This was a cross-sectional and retrospective cohort study with a descriptive, comparative, correlational, and multivariate analysis strategy. A convenience sample was recruited and were stratified into 2 performance level groups (talented vs. untalented) for each chronological age group consistent with their respective sports category (8.0–9.9 years, 10.0–11.9 years, 12.0–13.9 years). Twelve independent variables were assessed: chronological age—AGE—(years), height—HT—(cm), body mass—BM—(kg), body mass index—BMI—(kg/m^2^), squat jump—SJ—(cm), counter movement jump—CMJ—(cm), CMJ Abalakov—ABK—(cm), sprint 10 m—SP10—(s), sprint 30 m—SP30—(s), “repeated side-step”—RSS—(n), sit & reach test—SRT—(cm), forced vital capacity—FVC—(L). A performance ranking was established as the dependent variable for the correlation and multivariate analyses (multiple linear regression analysis).

### 2.2. Subjects

Data used in this study were collected in the 2013–14 midseason (late February to May 2014) and corresponded to 722 male Chinese children aged from 8.0 to 13.9 years, from 6 competitive categories and fifty-nine regional teams. Children pertained to the “Evergrande Football School” –EFS– (Qingyuan City, Guang Dong Province, China), a private school with a collaboration agreement with the Real Madrid Foundation (Real Madrid Football Club, Madrid, Spain), for the sports training and holistic education of soccer children. According to the expert Spanish coaches of EFS and following their technical and tactical criterion, based on the preview of a series of matches, 204 children were considered as talented (8.0–9.9 years: *n* = 41; 10.0–11.9 years: *n* = 114; 12.0–13.9 years: *n* = 49) while 518 as untalented (8.0–9.9 years: *n* = 150; 10.0–11.9 years: *n* = 232; 12.0–13.9 years: *n* = 136). This classification into 2 performance level groups for each competitive category conformed to a standard procedure commonly used in the EFS at the beginning of each sport season (the list of technical and tactical performance indicators can be consulted as Appendix A). Independently of this categorization, the authors of this study created a competitive ranking based on the previous league results of every team. The formula was: “final points achieved · 100/maximum possible points”. Thus, each player, grouped according to their chronological age, obtained a scoring scale that was comparable to the rest of the sample. The eligibility criteria were as follows: (1) male EFS soccer players aged between 8.0 to 13.9 who performed all the tests with a minimum of 2 recorded repetitions; (2) to have competed in the regional soccer league during the past season; (3) to be free of previous injuries and/or illnesses that may have affected the test results. From a total of 757 players, 35 players were finally discarded for not meeting the inclusion criteria.

### 2.3. Ethical Issues

All the assessments corresponded to measures or tests commonly used during soccer training sessions and whose rights to data processing or assignment belonged to EFS. Consequently, the study had the signed approval of the person legally responsible for the EFS. The study was conducted following the Helsinki Declaration Statement [30]. Retrospective protocol and procedures were approved by the Ethics Committee for Clinical Sport Research of Catalonia (Ethical Approval Code: 19/CEICGC/2020).

### 2.4. Procedures

All tests were performed by a single investigator according to previously standardized protocols. He was responsible of the physical conditioning area of EFS in the age groups analyzed and is the second author of this study. Anthropometric measurements and physical fitness tests were performed over 14 weeks, corresponding to the middle of the 2013–14 season, after the Chinese New Year (from the end of February to the end of May 2014). Every morning from Monday to Friday, from 9:30 a.m. to approximately 12:30 p.m., a whole team (12–14 subjects) performed all the tests in the EFS gym. This facility had a controlled temperature, set at 24.0 ± 1.0 °C, with a relative humidity of 55.5 ± 10.0%. Upon arriving at the gym, in an adjoining room, players were measured and weighed by the medical doctor responsible for the EFS, who also performed the spirometry test. Next, and even though all the players were accustomed with the tests, because they had previously performed them during training, they were reminded of protocols before the start. Then, all subjects performed a general warm-up (10–15 min at 60% heart rate) and a specific warm-up (10–15 min, including performing each of the tests as a pre-test). Except for the SRT, which was always performed as the last test, the execution order of the rest (SJ, CMJ, ABK, SP10, SP30, and SST) was randomized. Recovery time between tests was always complete (>3 min) and for each evaluation 2 attempts were performed to ensure the reliability of the measurements. Once verified, the best attempt was the one finally registered.

Specific protocols and materials used for every assessment are detailed as follows: Anthropometric measurements were performed according to the standard criteria of the International Society for the Advancement of Kinanthropometry [31]. HT was assessed to the nearest 0.1 cm using a telescopic stadiometer (Seca 220^®^, Hamburg, Germany), and BM was measured to the nearest 0.05 kg using a calibrated weighing scale (Seca 710^®^, Hamburg, Germany). BMI (kg/m^2^) was derived from BM/HT^2^. FVC was registered with a battery-operated portable spirometer (FCS-10000^®^; Grows Instrument Ltd., Hong Kong, China), following the previous protocols of the European Respiratory Society [32] and the recommendations on forced exhalation time in young adolescents, which were set to 6 s [33]. Each physical fitness test included in the assessment fulfilled with previously published standards and validated measurement protocols for children and adolescents: SJ, CMJ, and ABK [34,35] were assessed with Chronojump Boscosystem^®^, composed of a contact platform and its corresponding software interface, “Chronojump v1.3.9” (Chronojump, Boscosystem^®^, Barcelona, Spain). SP10 and SP30 followed previous methodological recommendations [36] and validated protocols for assessing youth sprint ability [37]. Both tests were automatically measured with 2 photoelectric barriers (Chronojump, Boscosystem^®^, Barcelona, Spain). SRT was performed according to the methodology of Eurofit [38] once its criterion-related validity had been verified for estimating the hamstring flexibility in children and adolescents [39]. Finally, regarding the RSS, despite being a widely used test in Asian schools and sports clubs to assess agility [40], no studies have been found on its validity prior to 2014, corresponding to the data of the present investigation. Recently, a research group from the Shanghai Sports University has published its protocol obtaining a strong concurrent validity when comparing its results against other previously published agility and movement skill tests [41]. Basically, RSS requires participants to demonstrate as many as possible repeated sideward steps in 20 s between 2 lines located 1 m apart (50 cm from the center, which would correspond to the starting position). The outcome measure is the total precise steps.

### 2.5. Statistical Analysis

Data are expressed by means and standard deviations. Normality and equal variance of the distributions were confirmed by the Kolmogorov-Smirnov and Levene tests, respectively. Test–retest reliability was examined using the intraclass correlation coefficient (ICC) with a two-way mixed average measures model. The coefficient of variation (CV) was calculated for all tests to determine the stability of measurement among trials with the 95% confidence interval. The Student’s unpaired t-test was performed to analyze possible differences between talented and untalented groups. For all those that were statistically significant, the effect size (ES) was calculated using Cohen’s d test. In order to check the differences between age groups, a one-way ANOVA was performed using the Tukey post-hoc test. The Pearson’s correlation coefficient was calculated to assess the relationship of each independent variable with the performance level (competitive results ranking). Finally, a stepwise multiple linear regression analysis (MLR) was performed to assess the ability of independent variables to explain the variance of the performance level. Precise p values were reported and *p* ≤ 0.05 was considered significant. All data were analyzed using SPSS 22^®^ (IBM Inc., New York, NY, USA). Effect size was calculated with G*Power v3.1.9.2 package (University of Kiel, Kiel, Germany).

## 3. Results

Test–retest reliability results are shown in Table 1. Anthropometric outcomes (HT, BM, BMI) did not differ in practically all 722 cases between the first and second measurements, hence we report the maximum reliability results. Regarding to the FVC and all physical fitness tests, high test–retest reliability was obtained (ICC values ranging from 0.77 to 0.97, and CV from 1.5 to 11.0% in both groups).

Results of the whole sample (*n* = 722) and statistical differences between talented (*n* = 204) and untalented players (*n* = 518), are shown in Table 2. No significant differences were recorded in AGE, FVC, or in any anthropometric variables (HT, BM, BMI). In contrast, both in the jumping tests (SJ, CMJ, ABK) and in the sprint tests (SP10, SP30), talented players showed significantly better performance than their counterparts (*p* ≤ 0.05), although with a small effect size (*d* ≤ 0.25). The largest difference was registered in the RSS test (*p* = 0.001), again in favor of the talented group, with a medium effect size magnitude (*d* = 0.49). Finally, SRT values were higher in the talented group, although these were not statistically significant (*p* = 0.39).

Differences for each age group and performance level are shown in Table 3. In the youngest group (8.0–9.9 years) there were no significant differences in any of the assessments performed between talented and non-talented players. In the 10.0–11.9 age group, talented players were significantly lighter and, consequently, had a lower BMI. They reported better performance in the CMJ, ABK, SP10, SP30, and RSS (*p* ≤ 0.03), although in all cases the effect size of these differences was small (*d* = 0.01–0.33), with the exception of RSS, in which it was medium (*d* = 0.51). The oldest talented group (12.0–13.9 years) only registered significantly higher values than their untalented counterparts in FVC (*p* = 0.05; *d* = 0.32) and RSS (*p* = 0.001; *d* = 0.83). On the other hand, when analyzing the differences between the 3 age groups, ANOVA results show that, regardless of the performance group (talented or untalented), players older than the preceding age were significantly heavier and taller, and they had a better performance in all tests performed. However, in SRT, no significant differences were found between younger and intermediate age groups, or between 8 years and 12 year-old players.

Statistically significant correlations between the competitive ranking and each of the variables analyzed were scarce and registered low (r = 0.3 to 0.5 or −0.3 to −0.5) or negligible values (r < 0.3 or −0.3). In the youngest age group, competitive performance of talented players seemed to maintain a certain relationship with their height (r = 0.36; *p* = 0.02) and their ability to cover 30 m in the shortest time possible (r = −0.31; *p* = 0.05). While in this age group (8.0–9.9 years) no correlation was registered in the untalented group, in the following group (10.0–11.9 years), older children (r = 0.18; *p* = 0.01) with a greater stature (r = 0.20; *p* = 0.003), a lower BM (r = −0.18; *p* = 0.01), and a greater lung capacity (r = 0.16; *p* = 0.01) seemed to obtain better competitive performance. This did not occur in talented players in this intermediate age group, where only lung capacity seemed to show a positive relationship with their competitive performance (r = 0.19; *p* = 0.05). In the oldest talented group (12.0–13.9 years), chronological age (r = 0.28; *p* = 0.05) and better results in SP30 (r = −0.31; *p* = 0.03) and RSS (r = 0.30; *p* = 0.04) seemed to be related to their position in the competitive ranking. Also, the ability to sprint, but this time in the SP10 (r = −0.19; *p* = 0.03), and the agility manifested in the RSS (r = 0.24; *p* = 0.01), were slightly related to performance of the least talented players in this age group. Multiple linear regression analysis reinforces the results of these correlations, with a low explanation of the variance of competitive performance (R^2^ ranged between 5.0 to 20.0%) in all age groups and independently of their classification as talented or untalented (Table 4).

## 4. Discussion

The purpose of this study was to provide relevant information on the basic anthropometric characteristics and some physical fitness tests commonly used in sports training in a large sample of young Chinese soccer players. To our knowledge, this is the first time that a multivariate model of competitive performance in young Chinese soccer players has been applied. The major finding of this study has been to verify the null or scarce relationship between anthropometric measures and/or physical fitness tests that coaches normally use during training sessions, and the classification of each player in a ranking based on their competitive league results as members of their respective teams.

Sport-specific technical skills assessments have been demonstrated a great sensitivity to discriminate different competitive levels and predict future sports performance in the talent identification area [42]. The present multidimensional proposal is based on a MLR analysis and has been previously applied to predict or explain sports performance, both in individual sports, such as swimming [43], triathlons [44], or artistic gymnastics [45], and in team sports, such as field hockey [46], handball [17,47], volleyball [48], or soccer [49,50,51]. However, its practical application in team sports is more complex, due to the necessary configuration of a ranking which acts as a response or dependent variable in MLR analysis, and which in many individual sports is already established by its own rules and regulations (e.g., ATP ranking in tennis, FINA points in swimming, ITRA points in trail running, etc.). In soccer, as well as in other team sports, the individual ranking must be configured based on quantitative parameters of game analysis (scouting reports) or by the criteria of expert coaches [15]. In this research, we have applied an individual score to each player based on the competitive results that they obtained as members of their respective teams. For example, the players of the team that was first classified in their respective league all obtained the same score. Although this criterion has certain limitations, because it does not consider individual intra-team differences, the expert coaches of the EFS considered it adequate because it was based on the results of a whole sport season and at the end of this period, in a sample of 722 players, this ranking differentiated between highest and lowest competitive level players.

Before discussing findings related to the rest of motor tests, it is important to note that sports performance is a complex process that involves many factors, including ethnic and racial differences [52,53]. Talent identification studies must consider cross-cultural or country analysis when attempting to identify—or to compare with other studies—the sports performance attributes [54]. Regarding Asian soccer players, racial differences have been found in the anthropometric and physiological profile of sixteen young Chinese elite male soccer players (16.2 ± 0.6 years) that, when compared to European and African counterparts, registered shorter stature and lower CMJ and SP30 values, among other physical fitness differences [28]. Comparing our results with those of other studies conducted with Chinese soccer players of similar ages is difficult, due to the lack of scientific information published in English. Only one study has been found that describes and correlates, by playing positions, some anthropometric and physical fitness attributes of seventy Chinese soccer players under 14 years [26]. According to our results in the 12.0–13.9 year-old players, HT (talented: 158.1 ± 9.9 cm, untalented: 157.1 ± 8.6 cm; *p* > 0.05) and BM (talented: 46.2 ± 9.0 kg, untalented: 44.8 ± 8.8 kg; *p* > 0.05) are comparable to their forwards soccer players (HT: 1.56 ± 0.11 m, BM: 43.9 ± 9.5 kg), being shorter and lighter than the rest of the players from other playing positions. As for fitness tests, average CMJ results, achieved in a study by Wong et al., were always greater than 50 cm, which were remarkable results for boys under 14 years, and therefore far from the 32.2 ± 5.3 cm (talented) and 31.1 ± 6.0 cm (untalented) values attained in our study. SP30 values are comparable again, as they stand between 4.81 ± 0.36 s (defenders) and 4.96 ± 0.4 s (forwards), and our results oscillate between 4.9 ± 0.4 s (talented) and 5.0 ± 0.3 s (untalented). Lastly, the authors found significant relationships of SP30 with BM (r = −0.54), HT (r = −0.64), and BMI (r = −0.24). CMJ also correlated with HT (r = 0.36). They conclude that talent identification in young Chinese soccer players should take into account the anthropometric profile, since it is related to some physical fitness performances, although these should not be an absolute selection criteria, since long-term performance depends on many other factors not sufficiently investigated at present.

In our case, there are no anthropometric differences between players classified by coaches as talented or untalented. Furthermore, no anthropometric variable correlates or explains the variance of performance. Similarly, although talented players are significantly better in jumping and sprinting tests, these outcomes appear to have poor or no relation to their subsequent competitive performance. These results are consistent with the practice of not overvaluing anthropometric measures or physical skill tests as tools for sports talent identification [18]. Several studies conducted with young European elite soccer players also relativize the motor tests’ importance for the selection processes [55,56,57], including, specifically, the sprint and jump performances [58]. Although it seems to be confirmed that covering larger distances at a high speed is a key performance factor in highly trained prepubertal soccer players [59], most current, related literature conclude that the best training strategy in these early stages is to let the players play, and that it is better to give them a broad spectrum of fundamental motor skills rather than prioritize the specific conditioning capacities in the training sessions to a large extent [14,15,60]. Related to this, positioning, and deciding when to play, seem to be key factors for talent development in soccer [61] within some technical characteristics related with the dribbling [62,63] and the ball kicking speed [64].

The early development process of young talented athletes must necessarily include not only the evaluation of physical, physiological, and technical skill components, but also the psychological and sociological factors that longitudinally affect sports performance [20,65,66]. This holistic, multidisciplinary approach to talent identification is currently recognized and accepted by both the scientific community and by a large part of sports coaches [54]. However, its practical implementation implies that both scientists and coaches must come together to perform holistic models to predict sports performance, in order to optimize training and obtain the maximum competitive achievements in the medium and long term [54,67]. The expert eyes or “gut instinct” of coaches and stakeholders are particularly important, but not everything can depend on this [68,69,70]. For example, in our study, the selection criteria between talented and untalented players was subjectively pre-established by the EFS coaches, based on the visualization of a series of matches. This suggests that they predictably focused on something more than in the physical skills of the young players. However, the final result of this classification was contradictory, because it was precisely only the physical differences that discriminated one group from the other, and not the competitive performance finally demonstrated by both, as reported by the results of the multivariate model.

The youth soccer Chinese training system is constantly growing and optimizing. To continue moving forward, next steps should focus on optimizing procedures for the detection, recruitment, and final selection of those players with the best aptitudes, skills, and competencies associated with future competitive success. Our research is consistent with currently related literature about talent identification: it is necessary to multidimensionally assume the complexity of sports performance, understanding it as a longitudinal, dynamic, and specific process dependent on each situation and context [71,72]. The opinion, criterion and experience of coaches should be complemented with scientific analysis based on psychologically and sociologically validated predictors, together with anthropometric, physiological, physical, and technical–tactical tests, among others [70]. For example, regarding to psychological aspects and specifically those referring to perceptual cognitive predictors [73], decision making during a soccer match seemed to be the most relevant variable to explain the competitive performance of 127 U12 and U15 soccer players (*d* = 0.81). Concerning to personality-related factors, some significant predictor variables were: hope for success, fear of failure, self-esteem, self-efficacy [74]. Finally, sociological variables have been studied to a lesser extent, although parental support, socio-economic background, education, coach–child interaction, hours in practice, and cultural background seem to explain the variance in the performance of young elite football players, as reported in a recent systematic review [75].

This is absolutely necessary in order to avoid the dissonance noted in this study between the opinion of the coaches when they identify talented or untalented players and the poor or null relationship with the subsequent tests performed to evaluate their present and/or future competitive performance.

Finally, this is a retrospective study with a convenience-recruited sample. This research design has some limitations, such as the possible bias derived from the quality of the data during its registration [76], that the sample is not representative of the population [77], or specifically in this study, two remarkable aspects: that it would have been desirable to introduce: an inter- and intra-rater reliability analysis, and a somatic biological maturity indicator (i.e., prediction of age at peak height velocity). In this context, some methodological considerations must be considered in order to define the contributions and limitations of the present study: (1) a large sample was recruited, a circumstance that minimizes possible rare outcomes and gives strength to the results; (2) the second author of this research was responsible for the execution and supervision of all the tests, that were performed twice under strict execution of the previously validated protocols. Only the RSS protocol was not validated in 2014, but it has recently been published [41]; (3) it is not considered indispensable to perform an inter- or intra-rater reliability analysis for three reasons: (a) assessed tests are commonly used in sports training and have validated protocols for young people, (b) operators responsible for the tests’ evaluation had extensive experience in these and, (c) test–retest reliability reports very high consistency results; (4) lastly, although it would have been scientifically desirable to operate with a biological age indicator, the present study has classified the age groups according to the corresponding competitive categories, a circumstance that will facilitate the understanding and application of the results by soccer coaches of many Chinese sports clubs.

## 5. Conclusions

This retrospective study provides relevant information on some basic anthropometric characteristics and physical skills of a large sample of young Chinese soccer players between 8.0 and 13.9 years old. A multivariate model configured from real competitive results is proposed in a novel way. Some practical considerations for coaches, clubs and training soccer institutions should be noted: Chinese soccer players classified as talented by expert coaches have better physical skills than untalented ones. However, these differences are neither related nor determine the competitive performance of one group or the other. The Chinese institutions and soccer professionals should consider these results to relativize the importance of anthropometric measurements and physical fitness tests in talent identification and development areas.

## Figures and Tables

**Table 1 healthcare-10-00098-t001:** Test–retest reliability statistics for 722 children, talented and untalented Chinese soccer players.

Tests	Talented (*n* = 204)			Untalented (*n* = 518)		
	ICC		CV	ICC		CV
	r	CI 95%	%	r	CI 95%	%
HT (cm)	1.00	--	0.0	1.00	--	0.0
BM (kg)	1.00	--	0.0	1.00	--	0.0
BMI (kg/m^2^)	1.00	--	0.0	1.00	--	0.0
FVC (L)	0.83	0.77–0.87	11.0	0.96	0.95–0.97	10.8
SJ (cm)	0.88	0.84–0.91	7.9	0.92	0.91–0.94	9.9
CMJ (cm)	0.89	0.86–0.92	6.5	0.94	0.93–0.95	7.3
ABK (cm)	0.89	0.85–0.91	5.6	0.94	0.93–0.95	7.3
SP10 (s)	0.91	0.88–0.93	1.6	0.84	0.80–0.86	5.2
SP30 (s)	0.95	0.94–0.97	1.5	0.94	0.92–0.95	4.2
RSS (n)	0.90	0.87–0.93	6.2	0.93	0.91–0.94	8.2
SRT (cm)	0.96	0.95–0.97	8.5	0.97	0.96–0.97	9.4

HT: height, BM: body mass, BMI: body mass index, SJ: squat jump, CMJ: counter movement jump, ABK: Abalakov, SP10: sprint 10 m, SP30: sprint 30 m, RSS: repeated side-step, SRT: sit & reach test, FVC: forced vital capacity. ICC: intraclass correlation coefficient, CV: coefficient of variation, CI 95%: confidence interval.

**Table 2 healthcare-10-00098-t002:** Descriptive results and comparison between anthropometric and physical fitness tests for 722 talented and untalented Chinese children soccer players.

Tests	All (*n* = 722)	Talented (*n* = 204)	Untalented (*n* = 518)	*t*-Test (*p*)	ES (*d-Cohen*)
AGE (years)	11.0 ± 1.4	11.1 ± 1.4	10.9 ± 1.4	0.08	--
HT (cm)	146.9 ± 10.7	147.0 ± 11.1	146.8 ± 10.5	0.84	--
BM (kg)	37.6 ± 8.9	37.3 ± 8.7	37.6 ± 9.0	0.69	--
BMI (kg/m^2^)	17.2 ± 2.2	17.0 ± 1.9	17.2 ± 2.4	0.21	--
FVC (L)	2.7 ± 0.7	2.7 ± 0.7	2.7 ± 0.7	0.52	--
SJ (cm)	27.0 ± 5.1	27.6 ± 5.1	26.8 ± 5.1	0.05 *	0.16
CMJ (cm)	27.6 ± 5.2	28.3 ± 5.1	27.3 ± 5.2	0.02 *	0.19
ABK (cm)	31.8 ± 5.9	32.5 ± 6.1	31.5 ± 5.8	0.04 *	0.17
SP10 (s)	2.2 ± 0.1	2.1 ± 0.1	2.2 ± 0.1	0.01 *	0.22
SP30 (s)	5.4 ± 0.4	5.3 ± 0.4	5.4 ± 0.4	0.001 *	0.25
RSS (n)	40.0 ± 4.6	41.6 ± 4.5	39.4 ± 4.5	0.001 *	0.49
SRT (cm)	8.9 ± 5.0	9.2 ± 4.9	8.8 ± 5.0	0.39	--

AGE: chronological age, HT: height, BM: body mass, BMI: body mass index, SJ: squat jump, CMJ: counter movement jump, ABK: Abalakov, SP10: sprint 10 m, SP30: sprint 30 m, RSS: repeated side-step, SRT: sit & reach test, FVC: forced vital capacity. *t*-test (talented vs. untalented): *p*: * statistical significance at *p* ≤ 0.05. ES: effect size (*d-Cohen*).

**Table 3 healthcare-10-00098-t003:** Comparative analysis applied to 722 Chinese children soccer players: One-way ANOVA between age groups (results are showed in rows) and Student’s t tests (talented vs. untalented), with the effect size showed in columns for significant differences.

Tests		8.0–9.9 Years	10.0–11.9 Years	12.0–13.9 Years
AGE (years)	Talented	9.0 ± 0.6	11.1 ± 0.5 *	12.9 ± 0.7 ^‡†^
	Untalented	9.2 ± 0.5	11.0 ± 0.6 *	12.7 ± 0.6 ^‡†^
*p t*-test & (*d*-Cohen)		ns	ns	ns
HT (cm)	Talented	135.9 ± 6.6	146.3 ± 8.1 *	158.1 ± 9.9 ^‡†^
	Untalented	137.4 ± 6.5	146.9 ± 7.6 *	157.1 ± 8.6 ^‡†^
*p t*-test & (*d*-Cohen)		ns	ns	ns
BM (kg)	Talented	30.8 ± 6.3	35.9 ± 6.0 *	46.2 ± 9.0 ^‡†^
	Untalented	31.4 ± 6.2	37.4 ± 7.4 *	44.8 ± 8.8 ^‡†^
*p t*-test & (*d*-Cohen)		ns	0.05 (0.22)	ns
BMI (kg/m^2^)	Talented	16.5 ± 2.1	16.7 ± 1.5	18.3 ± 1.9 ^‡†^
	Untalented	16.5 ± 2.3	17.2 ± 2.3 *	18.0 ± 2.4 ^‡†^
*p t*-test & (*d*-Cohen)		ns	0.02 (0.26)	ns
FVC (L)	Talented	2.2 ± 0.5	2.6 ± 0.5 *	3.5 ± 0.8 ^‡†^
	Untalented	2.2 ± 0.4	2.7 ± 0.5 *	3.3 ± 0.7 ^‡†^
*p t*-test & (*d*-Cohen)		ns	ns	0.05 (0.32)
SJ (cm)	Talented	24.4 ± 3.2	27.0 ± 4.4 *	31.6 ± 5.4 ^‡†^
	Untalented	24.1 ± 3.8	26.2 ± 4.0 *	30.7 ± 5.7 ^‡†^
*p t*-test & (*d*-Cohen)		ns	ns	ns
CMJ (cm)	Talented	24.5 ± 3.3	27.9 ± 4.5 *	32.2 ± 5.3 ^‡†^
	Untalented	24.7 ± 3.8	26.8 ± 4.2 *	31.1 ± 6.0 ^‡†^
*p t*-test & (*d*-Cohen)		ns	0.02 (0.25)	ns
ABK (cm)	Talented	27.5 ± 3.3	32.0 ± 5.0 *	37.7 ± 6.2 ^‡†^
	Untalented	28.3 ± 4.2	30.8 ± 4.5 *	36.0 ± 6.6 ^‡†^
*p t*-test & (*d*-Cohen)		ns	0.03 (0.25)	ns
SP10 (s)	Talented	2.3 ± 0.1	2.2 ± 0.2 *	2.1 ± 0.1 ^‡†^
	Untalented	2.3 ± 0.1	2.2 ± 0.1 *	2.1 ± 0.1 ^‡†^
*p* t-test & (*d*-Cohen)		ns	0.01 (0.01)	ns
SP30 (s)	Talented	5.7 ± 0.3	5.3 ± 0.3 *	4.9 ± 0.4 ^‡†^
	Untalented	5.8 ± 0.4	5.4 ± 0.3 *	5.0 ± 0.3 ^‡†^
*p t*-test & (*d*-Cohen)		ns	0.01 (0.33)	ns
RSS (n)	Talented	37.1 ± 3.2	41.2 ± 3.2 *	46.2 ± 3.8 ^‡†^
	Untalented	35.9 ± 3.6	39.5 ± 3.5 *	43.0 ± 3.9 ^‡†^
*p t*-test & (*d*-Cohen)		ns	0.001 (0.51)	0.001 (0.83)
SRT (cm)	Talented	8.2 ± 3.7	8.7 ± 5.1	11.2 ± 4.8 ^‡†^
	Untalented	8.8 ± 4.2	8.4 ± 5.1	9.6 ± 5.5 ^†^
*p t*-test & (*d*-Cohen)		ns	ns	ns

AGE: chronological age, HT: height, BM: body mass, BMI: body mass index, SJ: squat jump, CMJ: counter movement jump, ABK: Abalakov, SP10: sprint 10 m, SP30: sprint 30 m, RSS: repeated side-step, SRT: sit & reach test, FVC: forced vital capacity, ns: not statistically significant. ANOVA: age groups statistical differences set at *p* ≤ 0.01: * (8.0–9.9 vs. 10.0–11.9); ^‡^ (8.0–9.9 vs. 12.0–13.9); ^†^ (10.0–11.9 vs. 12.0–13.9). *p t*-test: “Talented vs. Untalented” statistical differences at *p* ≤ 0.05. Effect Size (*d*-Cohen): ≈0.20 small effect; ≈0.50 medium; ≈0.80 large.

**Table 4 healthcare-10-00098-t004:** Stepwise multiple linear regression analysis by chronological age groups in talented and untalented Chinese children soccer players.

Age (Years)	Talented (T) Untalent. (U)	Explicative Equations	*F*	df_1_	df_2_	*p*	R^2^		SEE
							Exact	Adjust.	
8.0–9.9	T (*n* = 41)	−64.63 + (0.98 HT) − (2.09 BMI)	5.97	1	38	0.01	0.24	0.20	9.33
	U (*n* = 150)	No variables selected in multivariate model	--	--	--	--	--	--	--
10.0–11.9	T (*n* = 114)	29.35 + (0.01 FVC) − (0.47 BM)	4.70	1	111	0.01	0.08	0.06	8.90
	U (*n* = 232)	15.72 − (0.04 HT) − (0.80 SP30) − (0.15 CMJ) + (0.11 ABK)	6.46	1	227	0.01	0.10	0.09	1.13
12.0–12.9	T (*n* = 49)	156.29 − (17.89 SP30) − (0.72 ABK)	4.83	1	46	0.01	0.17	0.14	8.89
	U (*n* = 136)	2.48 + (0.07 RSS)	8.04	1	134	0.01	0.06	0.05	1.13

HT: height, BMI: body mass index, FVC: forced vital capacity, BM: body mass, SP30: sprint 30 m, CMJ: counter movement jump, ABK: Abalakov; RSS: repeated side-step test, ANOVA statistics: Fentry (*p* ≤ 0.05) to Fexit (*p* ≥ 0.10), R2: coefficient of determination, SEE: standard error of the estimate, *p*: level of significance (*p* ≤ 0.05).

## Data Availability

All data generated analyzed during the current study are available from the corresponding author on reasonable request.

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
