# Peer review of "Physical Fitness and Performance in Talented & Untalented Young Chinese Soccer Players"

_healthcare, 2022, doi:10.3390/healthcare10010098_

Round 1
Reviewer 1 Report
The authors compared the data of anthropometric and fitness tests between the talented and un-talented young soccer players. The topic is interested and this is a well-written paper. The outcomes will be helpful to the process of identifying sports talent. There are only a few comments to be addressed.
- Line 26: the word of ‘relativized’ is not very clear. Do you mean that the application of data of anthropometric and fitness tests in young soccer players should be cautious? Or not overvalue these data as the tools for sport talent identification?
- Line 99: it would be better to list these ‘technical and tactical criteria’ in the Method or as a supplementary document if there are too many items. More information about the performance ranking is needed, as it is the dependent variable for the following analyses.
- Line 122-125: it is not necessary to tell who did the measurements, only if the standard protocols were followed.
- Line 220-221: ‘between younger and older untalented group’ should be ‘between 8 yrs and 12 yrs’, according to the last row of Table 3.
- Line 284-292: the discussion about FVC and SRT is not necessary I think, please consider removing this section.
- In the Conclusion, the last sentence of ‘The Chinese institutions and soccer professionals should consider these results to relativize physical skill tests in talent identification and development areas’. Please consider re-writing this sentence. It is not a strong and clear conclusion.
Reviewer 2 Report
I believe the topic of the article to be of interest for the readership. The format and organization of the manuscript meet scientific requirements. The results are presented clearly and well documented.
Some results should be contextualized, to better highlight the difference between performance enhancement related to normal growth advantages, higher-level fitness as opposed to athletic talent.
Even if the quality of the language is generally good, the manuscript would benefit from a read over to adjust some sentence structures.

Author Response
Please see the attachmet.
